# Live imaging reveals the progenitors and cell dynamics of limb regeneration

Frederike Alwes[1,2], Camille Enjolras[1,2], Michalis Averof[1,2]*

[1]Institut de Génomique Fonctionnelle de Lyon (IGFL), École Normale Supérieure de Lyon, Lyon, France; [2]Centre National de la Recherche Scientifique (CNRS), France

**Abstract** Regeneration is a complex and dynamic process, mobilizing diverse cell types and remodelling tissues over long time periods. Tracking cell fate and behaviour during regeneration in active adult animals is especially challenging. Here, we establish continuous live imaging of leg regeneration at single-cell resolution in the crustacean *Parhyale hawaiensis*. By live recordings encompassing the first 4-5 days after amputation, we capture the cellular events that contribute to wound closure and morphogenesis of regenerating legs with unprecedented resolution and temporal detail. Using these recordings we are able to track cell lineages, to generate fate maps of the blastema and to identify the progenitors of regenerated epidermis. We find that there are no specialized stem cells for the epidermis. Most epidermal cells in the distal part of the leg stump proliferate, acquire new positional values and contribute to new segments in the regenerating leg.

## Introduction

Many animals have the ability to regenerate parts of their body after suffering a severe injury. Like embryonic development, regeneration is a complex process, mobilising large numbers of cells in a regulated fashion to restore the missing parts to their original form and function. However, compared with embryonic development, regeneration is more difficult to study because it occurs in species and life stages that are less genetically tractable and accessible for continuous observation. Live imaging is particularly challenging for two reasons: first, regeneration is a lengthy process, unfolding on a timescale of weeks, and second, it occurs in larval and adult stages when animals tend to be large and highly mobile. These factors severely limit our ability to observe regenerating organs continuously, at sufficient temporal and spatial resolution. As a result, until now most regenerative studies have had to painstakingly reconstruct the dynamics of regeneration from series of static images, captured at different stages and across several individuals (e.g. *Emmel, 1910*). Recently, progress has been made in imaging the behaviour of stem cells in the context of physiological tissue turnover, repair and regeneration (*Lo Celso et al., 2009*; *Xie et al., 2009*; *Eilken et al., 2009*; *Rompolas et al., 2012*; *Mateus et al., 2012*; *Ritsma et al., 2014*; *Bradshaw et al., 2015*; *Barbosa et al., 2015*; *Webster et al., 2016*; *Gurevich et al., 2016*), but in all cases continuous imaging over long timescales has been challenging.

We recently introduced the small amphipod crustacean *Parhyale hawaiensis* as an experimental model for regenerative studies (*Konstantinides and Averof, 2014*) (*Figure 1A*), joining the ranks of previously established models such as fish, amphibians, flatworms and hydrozoans (*Goss, 1969*; *Reddien et al., 2004*; *Plickert et al., 2012*; *Gemberling et al., 2013*). Adult *Parhyale* can fully regenerate their limbs throughout their lifetime. A range of genetic approaches have been established in this species, including transgenesis, CRISPR-mediated gene editing, gene knockdown, gene mis-expression, mosaic analysis and gene trapping (*Pavlopoulos and Averof, 2005*; *Pavlopoulos et al., 2009*; *Liubicich et al., 2009*; *Ozhan-Kizil et al., 2009*; *Kontarakis et al., 2011*; *Konstantinides and Averof, 2014*; *Serano et al., 2016*; *Martin et al., 2016*). Genomic resources

*For correspondence: michalis.
averof@ens-lyon.fr

Competing interests: The authors declare that no competing interests exist.

include comprehensive transcriptomes and a draft assembly of the *Parhyale* genome (http://www. ncbi.nlm.nih.gov/genome/15533). Using these tools we started to investigate the process of limb regeneration in *Parhyale* (*Konstantinides and Averof, 2014*). Using clonal markers, we traced the contribution of different cell lineages to regenerated limbs, demonstrating that regenerated tissues arise from separate ectodermal and mesodermal progenitors, which reside locally in the amputated limb (*Konstantinides and Averof, 2014*). In the mesoderm, we discovered a population of *Pax3/7*-expressing satellite-like cells that can serve as muscle progenitors during leg regeneration (*Konstantinides and Averof, 2014*).

*Parhyale* has a number of attributes that make it well suited for live imaging of regenerating limbs. First, limb regeneration in *Parhyale* is relatively rapid, requiring as little as one week for young adults to fully regenerate their legs. Second, the *Parhyale* exoskeleton (cuticle) is transparent and the limbs are less than 100 µm in diameter, allowing us to image with single-cell resolution through their entire thickness. Third, the chitinous exoskeleton provides a robust support for immobilizing the amputated limb, while protecting the underlying tissues; we can glue the exoskeleton to a solid support without influencing the regenerative process that occurs inside the limb stump. Finally, the transgenic tools that we have established in *Parhyale* allow us to label the cells of the limb using a range of genetically-encoded fluorescent reporters.

Here we develop a method for immobilizing the amputated legs of active (non-anaesthetized) individuals, which allows us to image regeneration at cellular resolution, continuously over several days (*Video 1*, based on *Konstantinides and Averof, 2014*). Using transgenic lines expressing fluorescent proteins localized to nuclei or cell membranes, we are able to track individual cells, to trace their cell lineage and to observe their dynamic behaviours during the course of leg regeneration (*Videos 2–10*). Based on live imaging and cell tracking, we describe distinct phases of regeneration, characterized by different cell behaviours, we identify the progenitor cells for the regenerated epidermis of the leg, and present fate maps relating the position of cell progenitors in the regenerating limb bud (blastema) to their ultimate fate in the patterned, regenerated leg. Our method also provides an opportunity to re-evaluate the centuries-old concepts of epimorphosis and morphallaxis (*Morgan, 1901*) based on a direct observation of cell fates.

## Results

### Imaging *Parhyale* leg regeneration

Confocal microscopy on fixed specimens reveals the basic organization of regenerating *Parhyale* legs. During the first 3 days post amputation, within the exoskeleton of the amputated leg we observe that cells are organized in two distinct layers: an outer epithelial layer and an inner mass of loosely arranged cells. By clonally labelling the ectoderm with EGFP (as described in *Konstantinides and Averof, 2014*) we can see that the outer layer consists of ectodermal cells whereas the inner cell mass contains mostly mesodermal cells and a few ectodermal cells (*Figure 1D–D'' and E*). The latter are likely to represent neurons and glia, as they are associated with nerves, stained using antibodies for acetylated-tubulin (*Figure 1D'''*). Muscles are no longer visible near the amputation site, but remain intact in more proximal parts of the leg (*Figure 1G*).

To study the dynamics of leg regeneration we turned to live imaging. A major challenge for achieving this was finding a way to immobilize regenerating legs under the microscope, while keeping the animals alive. *Parhyale* adults are active, needing to move their limbs to feed and to generate water currents that facilitate breathing; long-term anaesthesia or immobilising whole animals in a solid matrix were therefore unlikely to succeed. We chose to immobilize individual thoracic legs for imaging by glueing them to the surface of a coverslip, using surgical glue (2-octyl cyanoacrylate) (*Schwade, 2008*). This approach allows us to continuously image the immobilized leg, through the coverslip, while allowing other parts of the animal to move freely (*Figure 1A,B*, *Video 1*). Approximately 80% of animals survive this procedure and release themselves from the coverslip with fully regenerated legs during their subsequent moult.

To visualize individual cells and to track their behaviour we employed transgenes that express fluorescent proteins specifically localised to the nuclei and mitotic chromosomes of cells (H2B-EGFP) or to the cell membrane (lyn-tdTomato), under the ubiquitously-expressed heat-inducible *PhHS* promoter (*Pavlopoulos et al., 2009*). The T2A ribosome skipping sequence (*Trichas et al., 2008*)

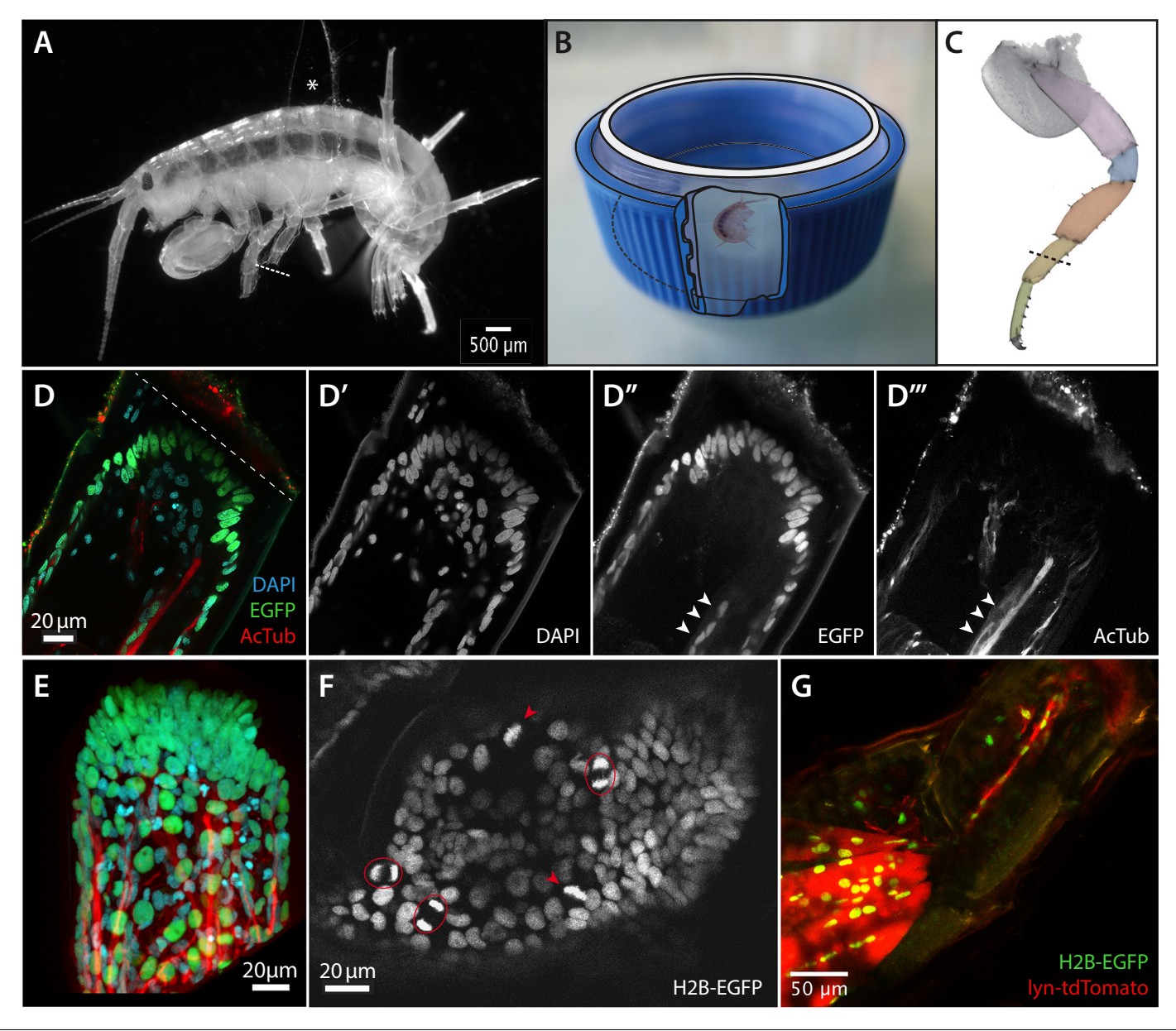

**Figure 1.** Imaging leg regeneration in *Parhyale hawaiensis*. (A) Amputated *Parhyale* adult mounted for imaging. The body of the animal is glued onto a coverslip, using a small piece of broken coverslip as a spacer (asterisk). The immobilized leg was amputated as marked with the dashed line. (B) Mounting of the coverslip carrying live *Parhyale* in a chamber for live imaging (see Materials and methods). (C) Outline of *Parhyale* thoracic leg (T4 or T5); individual podomeres are highlighted and the position of amputations marked with a dashed line. (D–D''') Cellular organization at the distal part of the amputated leg stump. Leg of a mosaic individual expressing H2B-EGFP specifically in the ectoderm (*Konstantinides and Averof, 2014*); fixed 63 hr post amputation and stained with antibodies for EGFP and acetylated tubulin to reveal ectodermal nuclei and neurons, respectively, and DAPI to label all nuclei. (E) 3-dimensional reconstruction of the same leg stump. (F) Single frame from live recording #04, showing histone-EGFP-labelled nuclei on the leg stump, 52 hr post amputation. Arrowheads and circles mark dividing cells in metaphase and telophase, respectively. (G) Leg stump of a mosaic individual expressing lyn-tdTomato and H2B-EGFP specifically in the mesoderm, 20 hr post amputation. Muscles persist in the proximal part of the leg stump but degenerate in the distal part (top right). The distal part of the leg stump contains a thin strand of interconnected mesodermal cells.

also allowed us to express both fluorescent proteins from the same transgene (*PhHS>lyn-tdTomato-2A-H2b-EGFP*). These transgenes allow us to track the position of each cell and to detect mitoses (*Figure 1F*, *Video 7*) and apoptotic events (seen as fragmenting nuclei, *Video 7*) throughout the

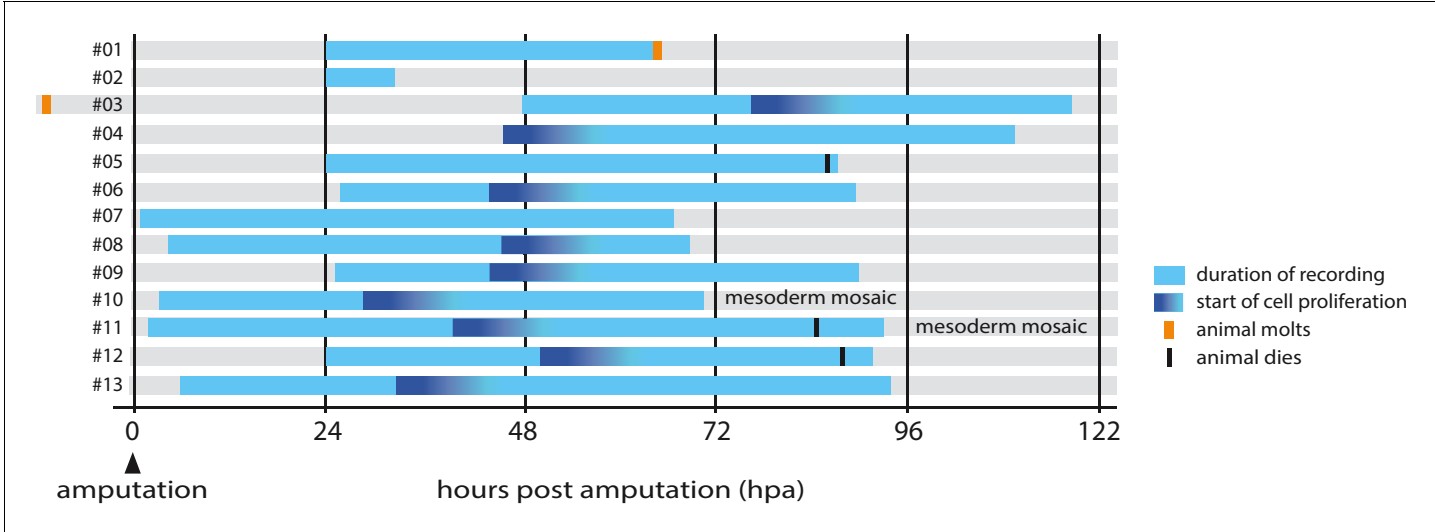

**Figure 2.** Overview of timelapse recordings captured by confocal microscopy. For each recording (numbered #1 to #13) we indicate the time period covered relative to the time of amputation (hpa, hours post amputation), the onset of cell proliferation, molting events and survival. Further details on each recording are given in *Table 1*.

leg, with single cell resolution. Haemocytes can be observed transiently, passing through the inner space of the leg (visible in single timelapse frames), and motile phagocytic cells ('macrophages') can

**Table 1.** Summary of live recordings by confocal microscopy.

| # | Transgenes | Start of recording (hpa) | Duration of recording (hours) | Timelapse interval (min) | Number of z slices | Step of z slices (μm) | Videos and Figures |
|---|---|---|---|---|---|---|---|
| 01 | PhHS>H2B-EGFP | 24 | 40 | 30 | 22 | 4.5 | |
| 02 | PhHS>H2B-EGFP | 24 | 8 | 15 | 25 | 4.6 | |
| 03 | PhHS>H2B-EGFP, DC5>DsRed | 48 | 63 | 45 | 22 | 3.1 | *Figure 5, 6A Videos 5, 6* |
| 04 | PhHS>H2B-EGFP | 46 | 64 | 30 | 28 | 2.1 | *Figures 1F, 6B Video 7* |
| 05 | PhHS>H2B-EGFP, DC5>DsRed | 24 | 62* | 30 | 19 | 3.5 | *Video 8* |
| 06 | PhHS>H2B-EGFP, DC5>DsRed | 26 | 63 | 30 | 16 | 2.5 | |
| 07 | PhHS>H2B-EGFP | 1 | 66 | 10, 15, 30 | 29 | 4.5 | *Video 4* |
| 08 | PhHS>lyn-tdTomato-2A-H2B-EGFP (unilateral ectoderm and mesoderm mosaic) | 5 | 65 | 30 | 22 | 4.5 | |
| 09 | PhHS>EGFP-PhGemN, PhHS>H2B-mRuby | 25 | 64 | 25 | 23 | 3.6 | |
| 10 | PhHS>lyn-tdTomato-2A-H2B-EGFP (mesoderm mosaic) | 3 | 68 | 30 | 21 | 3.5 | *Figure 1G* |
| 11 | PhHS>lyn-tdTomato-2A-H2B-EGFP (mesoderm mosaic) | 2 | 92* | 30 | 24 | 3.5 | *Video 10* |
| 12 | PhHS>H2B-EGFP, DC5>DsRed | 24 | 68* | 30 | 21 | 3.5 | |
| 13 | PhHS>EGFP-PhGemN, PhHS>H2B-mRuby | 6 | 88 | 30 | 20 | 3.8 | *Figure 6C Video 9* |

*Animal died during the recording (see *Figure 2*).

be seen engulfing cellular debris (*Video 8*). A separate transgenic line, *DC5>DsRed*, expressing DsRed specifically in neurons (*Konstantinides and Averof, 2014*), reveals that neurons are present in the regenerating leg stump at all stages of regeneration. Nerves can be seen retracting from the distal tip of the amputated leg while at the same time extending new processes towards it (*Video 6*).

Using these transgenic markers we captured 13 high-resolution recordings spanning the first 4.5 days of regeneration (summarized in *Figure 2* and *Table 1*).

## Distinct cell behaviours underpin different phases of leg regeneration

A specific sequence of events and cell behaviours unfolds during the course of regeneration at the distal 100–200 μm of the regenerating limb stump. Based on these events, we can subdivide the regenerative process into distinct phases (*Figure 3*). Each phase does not occur at precisely the same time in every individual, but the succession of phases is always the same.

The first event that takes place when a leg is amputated is wound closure (phase 1). Using two mosaic animals with EGFP-expressing haemocytes (with a *PhHS>EGFP* transgene integrated in the haemocyte lineage, but not in other cells of the amputated leg), we could observe bleeding immediately after leg amputation and very rapid closure of the wound through the adhesion of haemocytes to the wound surface (*Figure 4A–A''*, *Video 2*); these events take place within minutes after amputation.

Within the next 24 hr (phase 2), two processes contribute to further closing the wound. First, the surface of the wound becomes covered by a thick melanized scab (*Figure 4B–B''*, *Video 3*), a typical wounding reaction of arthropods mediated by blood cells (*Theopold et al., 2004*). Second, the epidermis of the leg migrates under the surface of the wound/scab, re-establishing an intact epithelial barrier between the scab and the inner tissues of the leg (*Figure 4B–B''*).

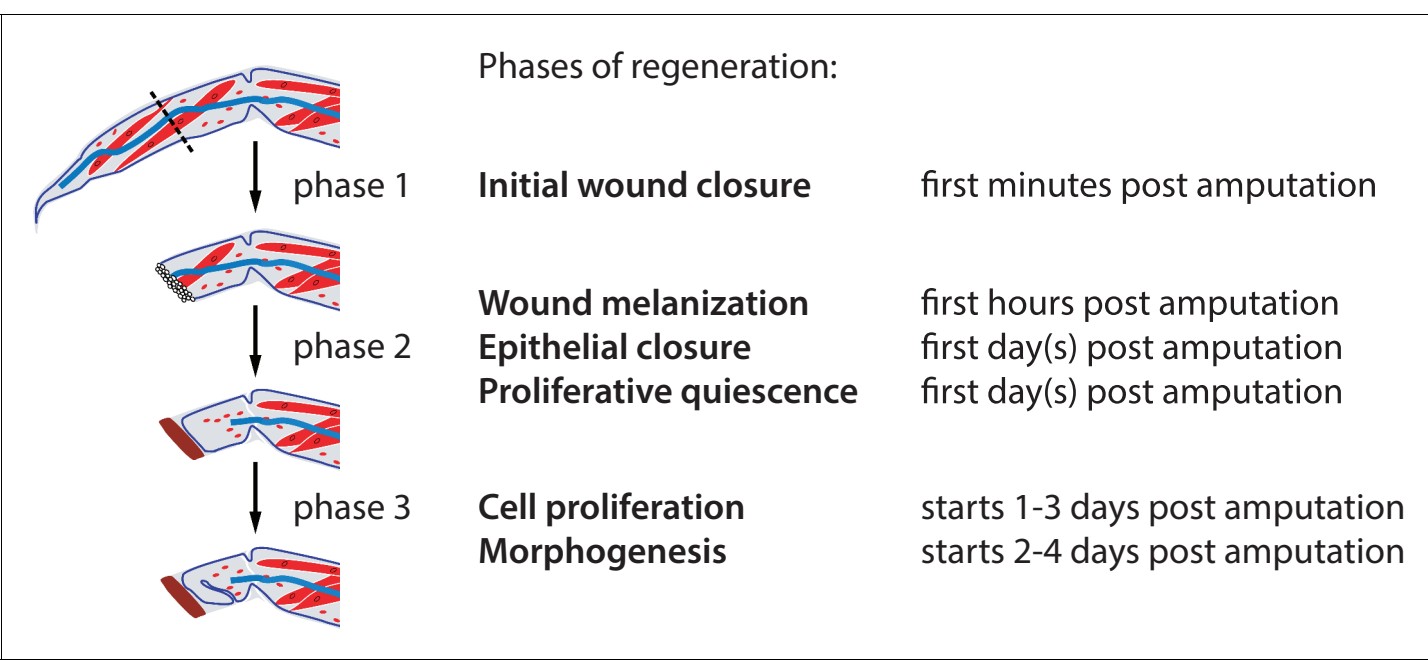

**Figure 3.** Phases of *Parhyale* leg regeneration. Three phases of regeneration are defined based on distinct cellular events and behaviours observed by live imaging (see text). Immediately after amputation, haemocytes adhere to the wound surface and close the wound (phase 1). In the hours that follow, a melanized scab (shown in brown) forms at the site of the wound, surrounding the haemocytes (early phase 2). The leg epithelium (depicted as a thin line surrounding the leg) then closes over the wound, under the surface of the melanized scab. Muscles (shown in red) at the distal part of the limb stump usually detach and degenerate, while those in proximal parts remain intact. During phase 2 we observe very limited or no cell proliferation. Phase 3 is marked by the onset of extensive cell proliferation and cell movements, leading to extensive growth and morphogenesis. This phase results in the formation of an elongated and patterned leg primordium within the amputated limb stump. Mesodermal cells, haemocytes and macrophages (shown in red), as well as nerves (in blue), are present in the inner spaces of the leg stump throughout the regenerative process.

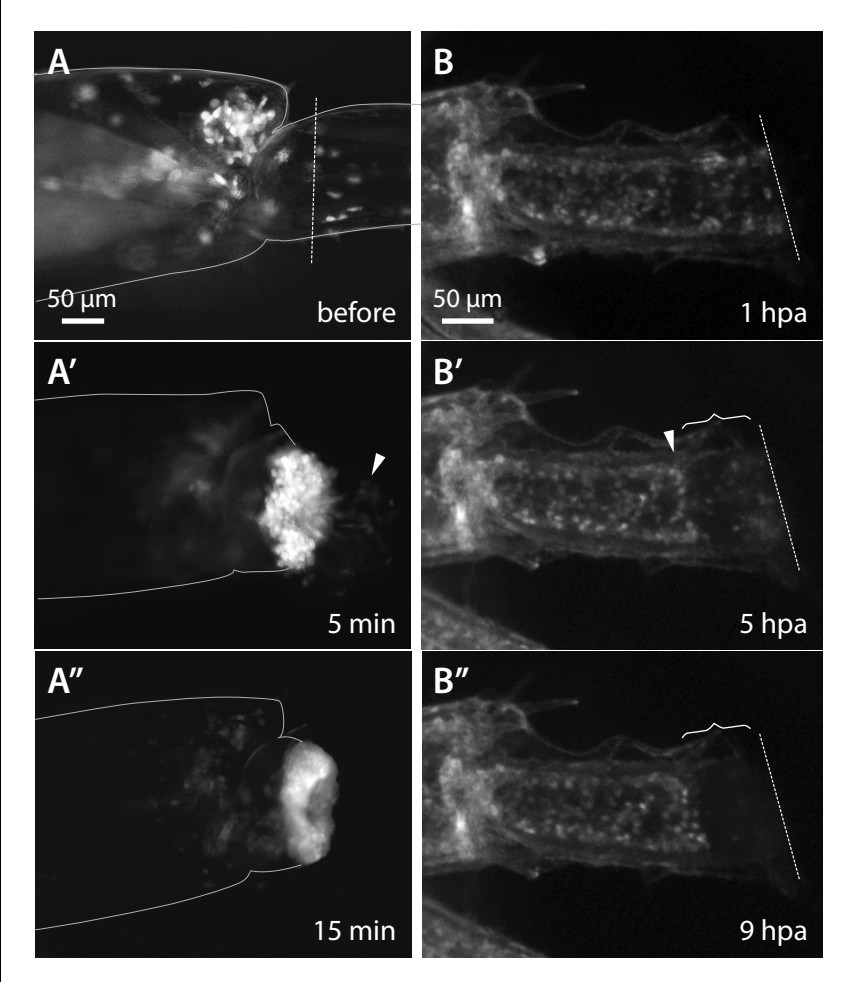

**Figure 4.** Wound closure during *Parhyale* leg regeneration. (A–A'') Mosaic *Parhyale* expressing EGFP in haemocytes show the role of these cells in the early stages of wound closure (same individual as in *Video 2*). (A) EGFP-labelled haemocytes in the leg prior to amputation; the outline of the leg and the amputation plane are highlighted. (A') In the first few minutes post amputation we observe bleeding (haemocytes marked by arrowhead) and haemocytes adhering to the wound. (A'') 15 min post amputation the bleeding has stopped and the wound is plugged by a mass of adhering haemocytes. (B–B'') Epithelial closure and wound melanization followed live in a transgenic animal expressing H2B-EGFP in all cells (still images from *Video 3*). (B) Amputated leg prior to epithelial closure and melanization; the amputation plane is marked with a dashed line. (B') 5 hr post amputation (hpa) a new epithelial layer has formed under the wound (arrowhead); more distally, the mass of haemocytes is not yet melanized (curly bracket). (B'') 9 hr post amputation, the distal part of the stump has been melanized; the haemocytes are embedded in the melanized scab and their fluorescence is no longer visible.

These events are followed by a 'quiescent' period, when epidermal cells continue to migrate slowly towards the site of the wound (*Figure 5B,B'*, *Video 4*) but no other activity can be seen under the microscope (late phase 2). During this period we observe almost no cell proliferation (*Video 4*), consistent with the very small number of EdU-incorporating cells detected during the first 1–2 days post amputation (*Konstantinides and Averof, 2014*).

This quiescent period is followed by a phase of extensive cell proliferation, cell movement and apoptosis (phase 3), starting around 1–3 days post amputation (*Figures 1F*, *2* and *5*, *Videos 5* and *7*). The transition is abrupt: within a few hours a large number of cells at the distal part of the stump start to divide (*Figure 5A'–A*). Based on our tracking of ectodermal cells, we estimate that the initial division rate is in the order of 0.4 cell divisions per cell per day, compared to <0.03 cell divisions per cell per day in the same cell lineages tracked during the quiescent phase (see Materials and

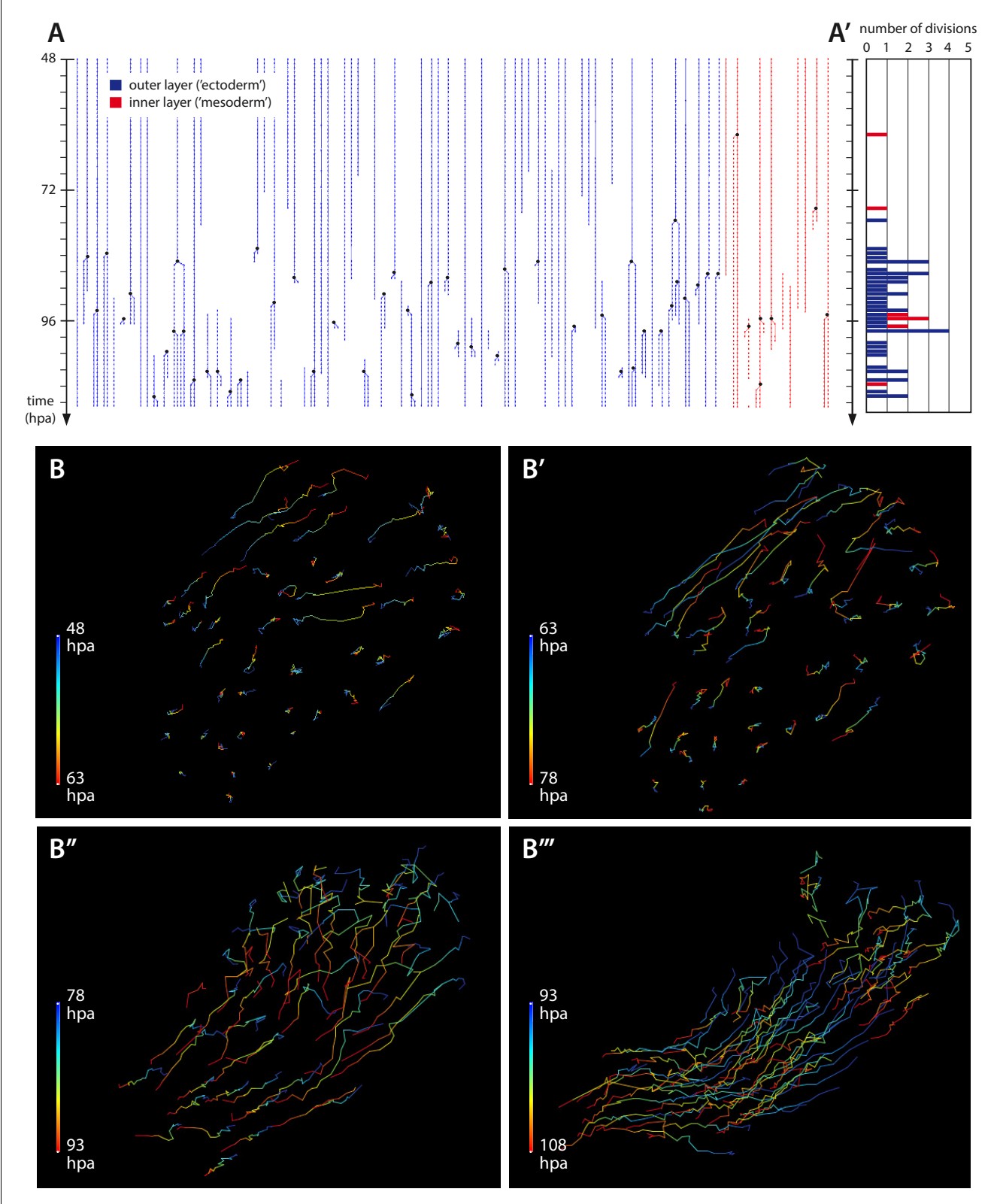

**Figure 5.** Cell proliferation and cell movements during *Parhyale* leg regeneration. (**A**) Cell lineages at the distal leg stump, tracked 48–111 hr post amputation (hpa) in recording #03. Vertical lines depict individual cells of the outer (blue) or inner (in red) cell layers tracked at successive timepoints. Cell divisions are depicted as lineage bifurcations and marked by a black dot. Incomplete lines indicate when a cell was not tracked throughout the recording. (**A′**) Histogram depicting the number of cell divisions observed at each timepoint. (**B–B′′′**) Tracks of individual epidermal cells during

*Figure 5 continued on next page*

*Figure 5 continued*

successive 15-hr time intervals, from recording #03 (*Video 5*), depicting cell movements at the distal leg stump. The amputation plane is at the top right corner of each panel. Each cell track is colour-coded such that the position of each cell at the start and the end of the given time interval are depicted in blue and red, respectively. (**B**, **B'**) Before the onset of extensive cell proliferation (48–78 hpa) cells show limited movements towards the wound surface or no movement. (**B''–B'''**) After the start of cell proliferation (78–108 hpa) epidermal cells participate in extensive morphogenetic movements. The length of the coloured bars corresponds to 50 microns.

methods). This initial burst of cell proliferation persists for approximately 20 hr, after which proliferation continues at lower levels (*Figures 5A,A'* and *7B*).

To visualize the transition from quiescence to cell proliferation we generated a cell-cycle reporter inspired by the vertebrate *Fucci* reporters (*Sakaue-Sawano et al., 2008*; *Sugiyama et al., 2009*). The reporter consists of EGFP fused to the N-terminus of *Parhyale* Geminin (PhGemN), marking cells at the S, G2 and M phases of the cell cycle when the EGFP-PhGemN fusion protein is stable. The fusion protein is rapidly degraded just after mitosis (see arrowhead in *Video 9*) due to cell-cycle-dependent degradation mediated by PhGemN. Recording #13 captures the initial burst of cell proliferation, involving a large number of cells at the distal end of the regenerating leg stump (*Video 9*).

Concurrent with cell proliferation, the regenerating leg also starts to display extensive morphogenetic movements (*Figure 5B''–B'''*): epidermal cells undergo complex cell rearrangements, which ultimately result in the formation of an elongated leg that becomes segmented (*Videos 5* and *7*).

Two mosaic animals, in which the *PhHS>lyn-tdTomato-2A-H2B-EGFP* transgene was specifically introduced in mesodermal lineages, allowed us to observe more specifically the behaviour of mesodermal cells (recordings #10 and #11). In the distal part of the limb stump we observe a population of mesenchymal cells that is particularly active during regeneration (*Video 10*). These cells are connected to each other, forming strands that run along the length of the leg stump, they proliferate and have active filopodial protrusions that extend in the inner space of the leg stump (*Video 10*).

## Fate maps and ectodermal progenitors

By tracking individual cells in our live recordings, we are able to trace the fate of cells during leg regeneration and to establish fate maps that connect the initial position of each cell in the regenerative bud (blastema) to its ultimate fate in the patterned leg.

Using mosaic animals in which the marker transgene has been incorporated specifically in mesodermal cell lineages (recordings #10 and #11) we observe that the inner and the outer cell layers of the leg stump do not mix during the course of regeneration. As expected from our previous mosaic analysis (*Konstantinides and Averof, 2014*), we did not observe any cells moving from the mesoderm to the epidermis.

Within the epidermal layer, we were able to track 54 cell lineages continuously in three independent recordings, over 25–64 hr periods, from the quiescent phase to segmented legs (recordings #03, #04 and #13, *Figure 6*). Many more lineages were partially tracked. We find that most of these cells divide and contribute small (2–8 cell) clones to the epithelium of the newly patterned leg, irrespective of their initial location in the epithelium of the blastema (*Figure 6A–C*). Thus, it appears that there is no specialized population of progenitor cells with the task of regenerating the epidermis; in *Parhyale* legs, most epidermal cells have the ability to serve as epidermal progenitors.

Next, we asked how the position of these cells in the epidermis of the limb stump relates to the position and contributions of their progeny in the patterned, regenerated leg. We find that the relative position of cells along the proximo-distal axis of the leg is roughly maintained from blastema to patterned leg. Although their spatial relationships become partly scrambled during morphogenesis, we observe that, on average, distally-located progenitors contribute to more distal portions of the regenerating leg than proximally-located progenitors (*Figure 6A'–C'*). In order to generate new distal segments, epidermal cells change their positional identity and become re-assigned to new leg segments (podomeres). For example, in recording #04, cells originating from the carpus of the amputated leg contribute both to the carpus and to the propodus of the regenerated leg (*Figure 6B*).

## Morphogenesis of the leg

Within two days from the onset of cell proliferation, the amputated limb stump is transformed into an elongated leg bearing the series of podomeres typical of *Parhyale* legs (see leg outlines in *Figure 6*). Our live recordings and cell tracking provide the first opportunity to investigate cell behaviours during this morphogenetic process. We focused on leg elongation. During the proliferative and morphogenetic phase of leg regeneration (phase 3, *Figure 3*), the length:width ratio of the proliferating epidermis increases by more than 2-fold. Individual cell elongation does not appear to be the basis for leg elongation (data not shown), but oriented cell division and/or cell rearrangements could be involved. To address this point, we directly measured the orientation of all epidermal cell divisions in recording #04. We find no evidence for a bias in the orientation of cell divisions, at any stage in the recording (*Figure 7A*). We therefore suggest that elongation is largely driven by cell rearrangements.

In most arthropods, regenerating limbs have to grow within a limited space, constrained by the unyielding exoskeleton that surrounds the amputated limb stump (*Goss, 1969*; *Maruzzo et al., 2013*). Our recordings provide some insights into how this is achieved. First, the epidermis detaches from the cuticle and the proximal, non-proliferating parts of the leg retract slowly towards the body wall, making space for the growth of more distal regenerating tissues (*Figure 7C*, *Video 7*). As the epidermis of proximal leg segments detaches, motile cells, likely phagocytes, can be seen transiently in the space between the exoskeleton and the epidermis (data not shown). Second, the regenerating

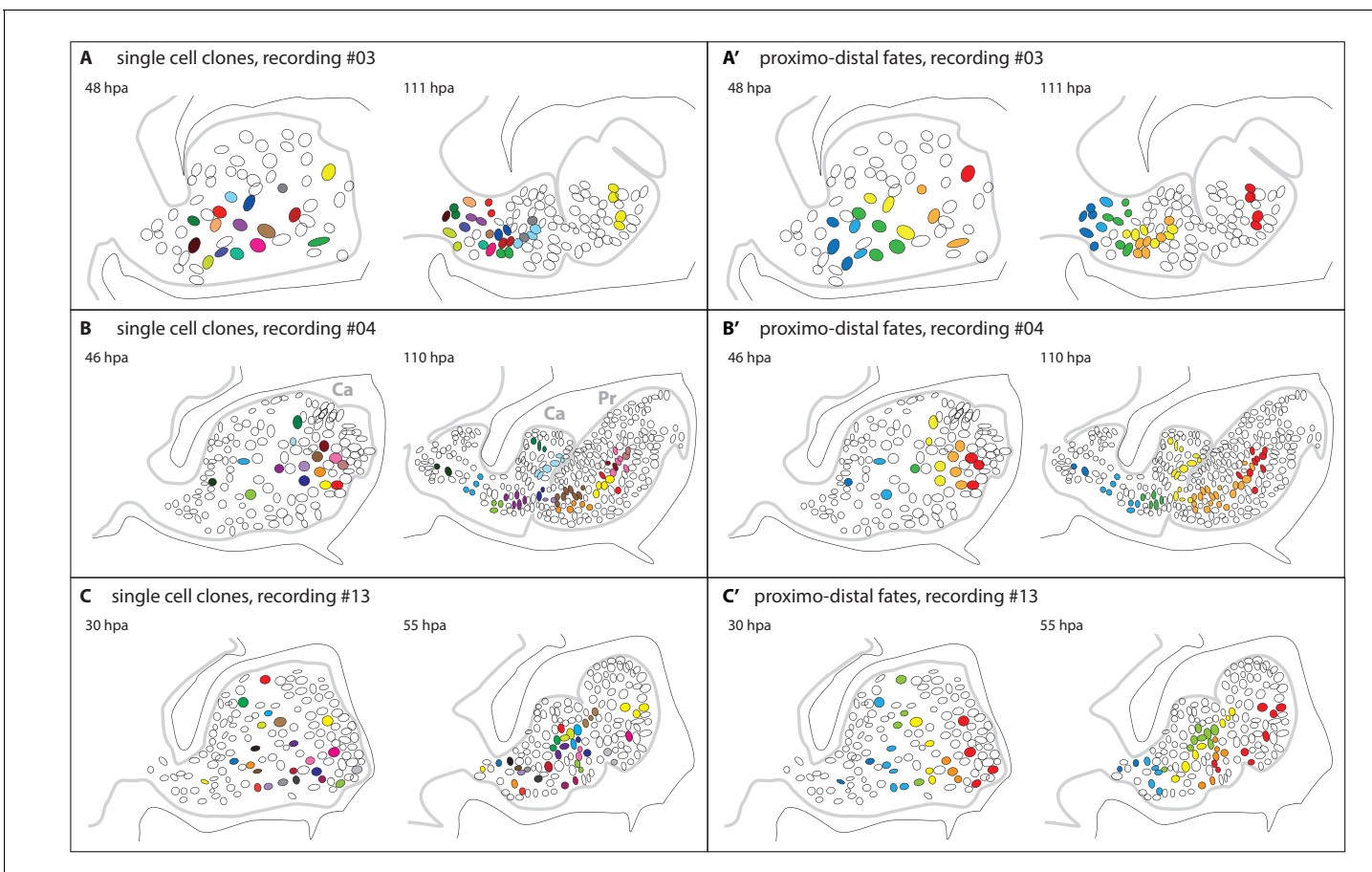

**Figure 6.** Epidermal cell clones and fate maps in regenerating leg stumps. (**A–C**) Tracked cells and their clonal progeny are shown from three separate recordings, #03, #04 and #13, respectively. The illustrations depict the outline of the regenerating leg epithelium (grey line) at the start (left) and the end (right) of cell tracking. Individual nuclei and their clonal progeny are colour coded. The leg stumps are shown with their distal end to the right and ventral side down. The outline of the surrounding exoskeleton is depicted by a black line. (**A'–C'**) Same tracking data, colour-coded according to the proximo-distal location of each nucleus at the start of cell tracking. In panel 6B, two podomeres, the carpus (Ca) and the propodus (Pr), are labelled.

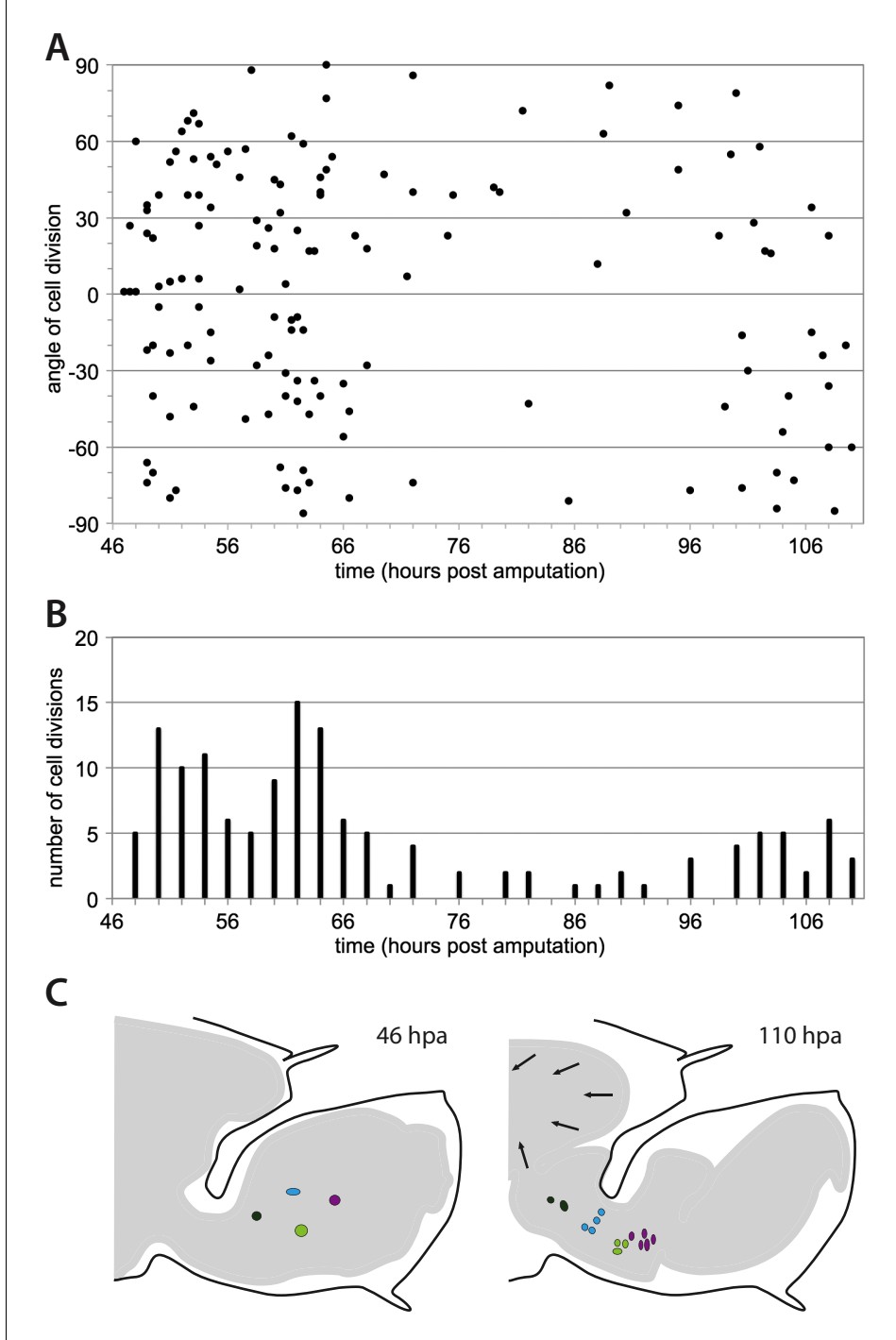

**Figure 7.** Morphogenesis of regenerating *Parhyale* legs. (**A**) Orientation of cell divisions occurring during the proliferative phase in recording #04 (46–110 hr post amputation). Each point represents a cell division; divisions occurring along the direction of the proximo-distal axis of the leg are at 0˚. (**B**) Number of cell divisions occurring in successive 2-hr time intervals, in the same dataset (recording #04). (**C**) Overall shape of regenerating leg before and after elongation (from recording #04). The outer black line shows the exoskeleton of the amputated leg that constrains the regenerating leg. Arrows highlight the detachment of the epidermis from the exoskeleton and the retraction of the leg proximally. Coloured circles depict the proximal movement of four cell clones (taken from *Figure 6B*) in the proximal part of the blastema.

leg becomes folded to accommodate the elongating proximo-distal axis into the available space (*Videos 5* and *7*). The legs do not immediately grow to their full size; they require additional moults to reach the size of equivalent non-amputated legs. For the type of amputations that we performed, legs reach 80–90% of the length of controls at the first moult after regeneration.

## Discussion

Our results show that *Parhyale* offers unprecedented opportunities for studying regeneration through live imaging at single-cell resolution. Live imaging has allowed us to follow the temporal dynamics of regeneration, capturing the behaviour of individual cells – in an organ that consists of a few hundred cells – over several days. The knowledge gained by this approach could not have been reached by imaging fixed samples.

First, we have gained an understanding of the temporal dynamics of leg regeneration. The most surprising observation is the sharp transition from a quiescent phase, during which there is hardly any cell proliferation, to a phase where the majority of cells at the distal part of the leg stump divide. This coordinated transition suggests that cells are responding to a common temporal signal. Since the onset of proliferation does not occur at a set time after amputation, but varies among individuals (*Figure 2*), we think it is unlikely to depend on a pre-set cell-autonomous timing mechanism. Rather, we favour the idea that it is triggered by a non-autonomous signal that coordinates this response among all the cells of the blastema. The nature and source of that signal are questions to probe in the future.

A second major gain from live imaging has been the opportunity to observe the behaviour of different cell types during regeneration. Notably, our recordings have allowed us to identify a previously unknown population of mesodermal cells, which proliferate and extend filopodial protrusions in the inner space of the blastema (*Video 10*). Though we are currently unable to determine whether these cells derive from satellite-like cells or from de-differentiating muscle fibres, we know that at least some satellite-like cells are present in the blastema during these stages (*Konstantinides and Averof, 2014*). The behaviour of these mesodermal cells resembles that of activated myogenic progenitors during muscle repair in mammals and zebrafish (*Webster et al., 2016*; *Gurevich et al., 2016*) and it is tempting to speculate that they too include the progenitors of regenerating muscle.

Live imaging has also allowed us to observe the complex behaviour (retraction and regrowth) of nerves and to visualize the movement of macrophages in the blastema (*Videos 6* and *8*).

Third, by single cell tracking we have been able to determine the cell genealogies and fates of more than 50 epidermal cells. We find that almost every epidermal cell that we tracked in the blastema contributes as a progenitor to the regenerated epidermis (*Figure 6*). This contrasts with our previous findings in the mesoderm, showing that muscle fibres arise from a set-aside population of stem cells (*Konstantinides and Averof, 2014*). Our current recordings do not allow us to trace the progenitors of other differentiated cell types, such as tendons, neurons and glia; tracing these would require longer recordings and markers for each of these cell types. But our live imaging method establishes a clear approach for unravelling these cell lineages in the future.

Cell tracking has also allowed us to investigate how the initial position of progenitor cells in the blastema relates to the position and contributions of their progeny in the regenerated limb. Based on the characteristic shapes of each leg segment, we are able to assign cells to specific podomeres of the leg. We observe that epidermal cell clones tend to preserve their relative positions along the proximo-distal axis of the leg. However, to re-establish the entire proximo-distal axis, cells need to be re-assigned to new podomeres (*Figure 6B*). This re-specification of positional identities brings to mind the old concept of morphallaxis.

The concepts of epimorphosis and morphallaxis have influenced regenerative studies for more than a century (*Morgan, 1901*). Morphallaxis describes a mode of regeneration where the regenerated tissues largely derive from re-patterning of existing cells, in the absence of cell proliferation – a mode exemplified by head regeneration in hydra. In contrast, epimorphosis describes a mode where regenerated tissues derive largely from new cells generated by cell proliferation in the blastema – exemplified by salamander limb regeneration. As noted by others, epimorphosis and morphallaxis are likely to represent extreme paradigms in a continuum that includes many intermediates (*Agata et al., 2007*; *Galliot and Ghila, 2010*). Our results demonstrate that *Parhyale* leg

regeneration represents an intermediate case where both re-patterning and growth (by cell proliferation) play important roles.

Finally, our work uncovers events that resemble processes occurring during regeneration in other phyla. The role of haemocytes in wound closure, the migration of macrophages towards the wound and apoptosis taking place in the blastema echo similar observations from other animals, where these processes play important roles in regeneration (*Bohn, 1975*; *Babcock et al., 2008*; *Godwin et al., 2013*; *Tseng et al., 2007*; *Chera et al., 2009*; *Li et al., 2010*; *Morata et al., 2011*). The continuous presence of neural projections in the blastema would also be consistent with a functional role of nerves in regeneration, as suggested by classic microsurgical experiments performed in diverse animals (*Singer, 1952*; *Kumar and Brockes, 2012*; *Needham, 1945*). In the future, it will be interesting to compare these processes across phyla in detail to pinpoint similarities and differences in the mechanisms of regeneration.

## Materials and methods

### Constructs, transgenic lines and mosaics

The *PhHS>H2B-EGFP* reporter, expressing the coding sequence of *Drosophila* histone 2B fused with EGFP under the *PhHS* cis-regulatory sequence (*Pavlopoulos et al., 2009*), was kindly provided by A. Pavlopoulos in plasmid pMi(3xP3>DsRed; PhHS>H2B-EGFP). A similar construct was used for histone 2B fused with mRuby (*PhHS>H2B-mRuby*).

To generate the *PhHS>lyn-tdTomato-2A-H2B-EGFP* reporter, we cloned the coding sequence of lyn-tdTomato from plasmid pCS2+_lyn-tdTomato (kindly provided by M. Handberg-Thorsager) in frame with the 2A-H2B-EGFP coding sequence from plasmid pCS2-TdTomato-2A-GFP (*Trichas et al., 2008*), downstream of the *PhHS* cis-regulatory sequence in plasmid pSL(PhHS-DsRed) (*Pavlopoulos et al., 2009*), replacing DsRed. The *PhHS>lyn-tdTomato-2A-H2B-EGFP* construct (see *Supplementary file 1*) was then cloned into pMi(3xP3>EGFP) (*Pavlopoulos et al., 2004*) using AscI, to generate plasmid pMi(3xP3>EGFP; PhHS>lyn-tdTomato-2A-H2B-EGFP).

To generate the *DC5>DsRed* reporter, we cloned an EcoRI fragment containing the *DC5* cis-regulatory sequence from pSLfa(DC5)GFPfa (*Blanco et al., 2005*) into a derivative of plasmid pSL(PhHS-DsRed) (*Pavlopoulos et al., 2009*) that lacks the first 845 nucleotides of the PhHS sequence (lacking the putative HSF binding sites but retaining the putative core promoter of *PhHS*, see *Kontarakis et al., 2011*). This combines the *DC5* element with the *PhHS* core promoter and the DsRed coding sequence (*DC5>DsRed*). The *DC5>DsRed* construct (see *Supplementary file 1*) was then cloned into pMi(3xP3-DsRed) (*Pavlopoulos and Averof, 2005*) using AscI, to generate plasmid pMi(3xP3>DsRed; DC5>DsRed).

To generate the cell cycle reporter, we identified a *Parhyale Geminin* orthologue in the *Parhyale* transcriptome and generated a construct where the 5' end of the *Parhyale Geminin* coding sequence (PhGemN, encoding amino acids 1–142) was placed downstream of the EGFP coding sequence (sequence accession KX130867) by gene synthesis (Biomatik, Canada). A NcoI-NotI fragment containing the EGFP-PhGemN sequence was then cloned downstream of the *PhHS* regulatory sequence in pSL(PhHS-DsRed) (*Pavlopoulos et al., 2009*), replacing DsRed. The *PhHS>EGFP-PhGemN* construct (see *Supplementary file 1*) was then cloned into pMi(3xP3-DsRed) (*Pavlopoulos and Averof, 2005*) using AscI, to generate plasmid pMi(3xP3>DsRed; PhHS>EGFP-PhGemN).

Plasmids pMi(3xP3>DsRed; PhHS>H2B-EGFP), pMi(3xP3>EGFP; PhHS>lyn-tdTomato-2A-H2B-EGFP), pMi(3xP3>DsRed; DC5>DsRed) and pMi(3xP3>DsRed; PhHS>EGFP-PhGemN) were microinjected into *Parhyale* embryos to generate stable transgenic lines and mosaics using established methods (*Pavlopoulos and Averof, 2005*; *Konstantinides and Averof, 2014*).

### Live imaging

For live imaging, adult animals were subjected to heat-shock for 45 min at 37°C to induce transgene expression via the *PhHS* promoter (*Pavlopoulos et al., 2009*) 24 hr before starting to record. To immobilize legs for live imaging we used surgical glue (2-octyl cyanoacrylate, Dermabond). The anesthetized animal was positioned laterally onto a round coverslip (Ø 30 mm #1, Thermo Fisher Scientific, USA) with its T4 and/or T5 legs glued onto the surface of the glass; a piece of broken coverslip was used as a spacer, glued to the body of the animal and to the round coverslip (*Figure 1A*,

*Video 1*). The animal was allowed to recover in artificial seawater for at least 2 hr prior to amputation. The immobilized T4 or T5 leg was then amputated at the carpus (position shown in *Figure 1C*) using a micro knife (Fine Science Tools GmbH, Germany). The coverslip bearing the animal was placed into the lid of a 50 ml Falcon tube where a large circular window had been cut out, fixed by screwing the top end of the Falcon tube and sealed with vaseline (*Figure 1B*). The resulting chamber was filled with ~3 ml of artificial seawater (specific gravity 1.018).

Imaging was performed on a cLSM Zeiss 780 inverted confocal microscope using a 20x objective (EC Plan-Neofluar) at ambient temperature (20–24°C). Z-stacks spanning the entire depth of the leg were captured at 2–5 μm steps, every 10–45 min with an image resolution of 1024 × 1024 pixels (see Table 1).

The images shown in *Figures 1A*, *4A''–A* and *Videos 1* and *2* were captured on a Zeiss Axio Zoom V16 microscope. The images shown in *Figure 4B–B''* and *Video 3* were captured on a Leica M205FA stereoscope; the still images were processed using the HeliconFocus software to combine images taken at different focal planes.

## Cell tracking, estimate of division rates

The timelapse recordings were cropped (and merged when necessary) in one multi-image TIFF hyperstack using Fiji (*Schindelin et al., 2012*). Nuclei were manually tracked in Fiji using the Track-Mate plug-in (http://fiji.sc/TrackMate).

To estimate the division rate of epidermal cells we counted the number of cell divisions occurring in tracked ectodermal cell lineages in recording #03 (*Figure 5A*) and measured the time sampled by these lineages in the quiescent and proliferative phases (48–81 and 81–111 hpa, respectively): in the quiescent phase we counted 1 cell division per 828 hr of cell tracking (0.03 divisions per cell per day), in the proliferative phase we counted 17 divisions per 999 hr of cell tracking (0.41 divisions per cell per day). The latter measurement is consistent with rough estimates of division rates derived from the total number of cell divisions observed in recording #04 (*Figure 7A–B,* 0.3 to 0.6 divisions per cell per day) and from the size of cell clones tracked in recordings #03, #04 and #13 (*Figure 6,* 0.3 to 0.7 divisions per cell per day).

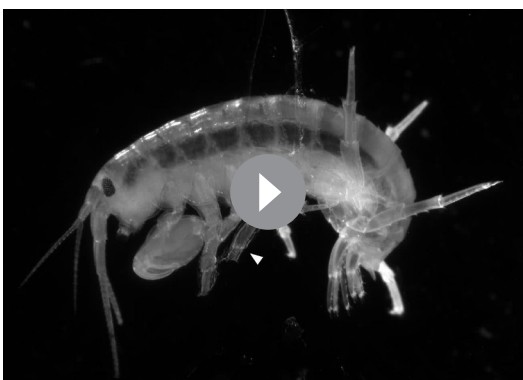

**Video 1.** Amputated *Parhyale* adult mounted for live imaging. Video of the individual shown in *Figure 1A*, moving extensively while an amputated leg remains immobilised on the coverslip. The amputated limb is marked by an arrowhead in the first frame of the movie.

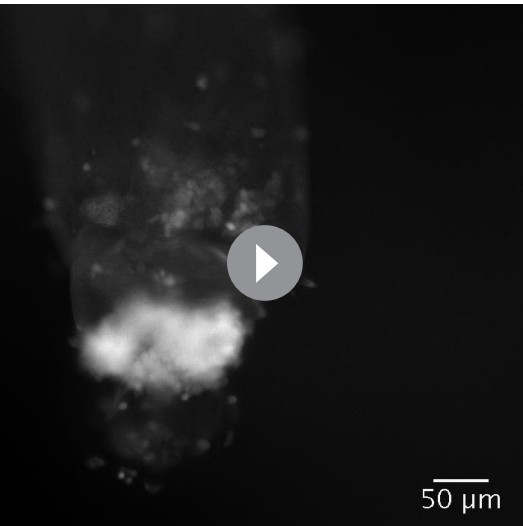

**Video 2.** Real-time imaging of amputated *Parhyale* leg, 5 min post amputation. This mosaic individual has an insertion of an EGFP-expressing transgene specifically in the Mav lineage, labelling haemocytes. We can observe bleeding and adherence of haemocytes to the wound surface. This individual was anaesthetised using clove oil and imaged without our usual mounting procedure.

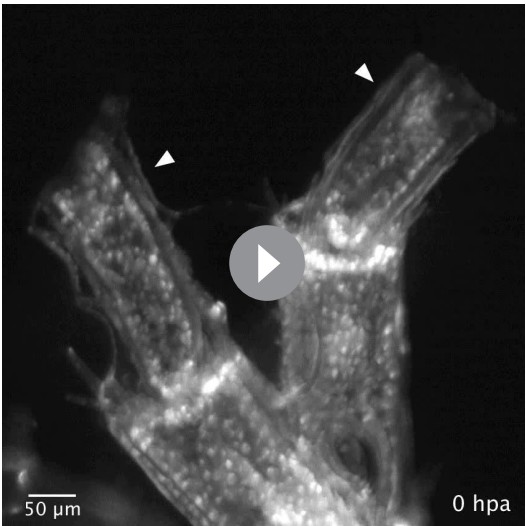

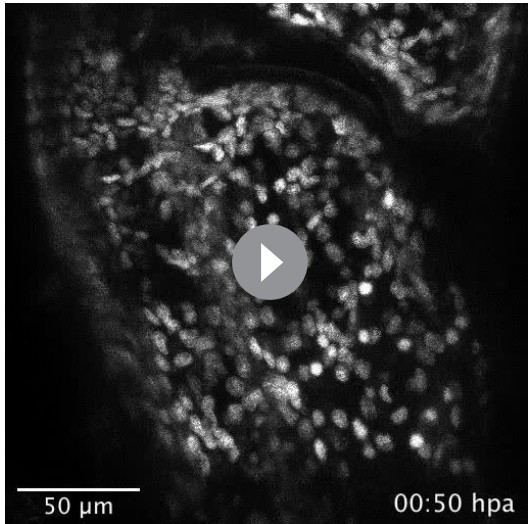

**Video 3.** Live imaging of two amputated *Parhyale* legs, 0 to 14 hr post amputation (hpa), using histone-EGFP to visualize all nuclei. Histone-EGFP is expressed from the *PhHS>H2B-EGFP* transgene after a heat shock. We can observe melanization of the wound at the distal end of each leg stump (arrowheads).

**Video 4.** Live imaging of amputated *Parhyale* leg, 1 to 67 hr post amputation (hpa), using histone-EGFP to visualize all nuclei. Histone-EGFP is expressed from the *PhHS>H2B-EGFP* transgene after heat shock. Maximum projection of focal planes capturing the surface of the leg epithelium, from recording #07. We can observe the rapid motility of some cells, probably macrophages, and the slower movement of epithelial cells towards the wound site, located at the bottom of the frame (~15–40 hpa). The video was assembled from three separate clips (0:50–3:50, 4:20–18:20 and 18:55–66:55 hpa) captured with different settings.

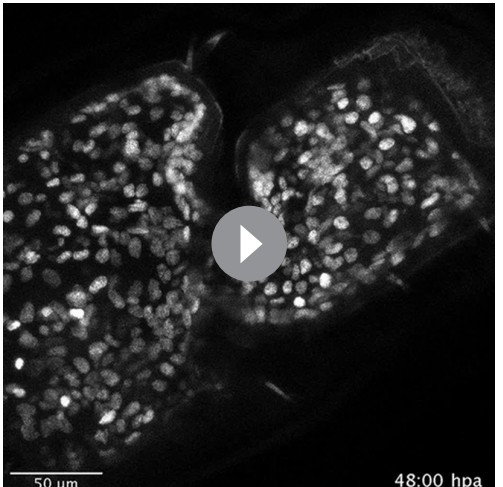

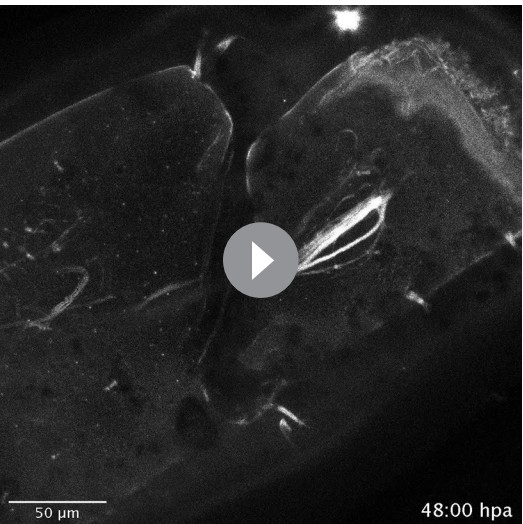

**Video 5.** Live imaging of regenerating *Parhyale* leg, 48 to 111 hr post amputation (hpa), using histone-EGFP to visualize all nuclei. Histone-EGFP is expressed from the *PhHS>H2B-EGFP* transgene after a heat shock. Maximum projection of green fluorescence channel, from recording #03. The amputation plane is at the top-right corner of the frame.

**Video 6.** Live imaging of regenerating *Parhyale* leg, 48 to 111 hr post amputation (hpa), using DsRed to visualize neurons. DsRed is expressed from the *DC5>DsRed* transgene. Maximum projection of red fluorescence channel, from recording #03 (same recording as *Video 2*).

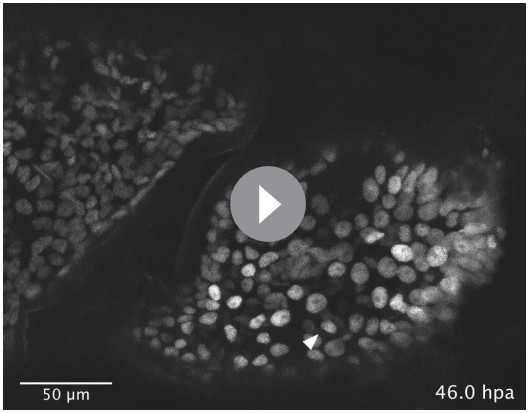

**Video 7.** Live imaging of regenerating *Parhyale* leg, 46 to 110 hr post amputation (hpa), using histone-EGFP to visualize all nuclei. Histone-EGFP is expressed from the *PhHS>H2B-EGFP* transgene after a heat shock. Arrowheads highlight two apoptotic nuclei. Maximum projection of focal planes capturing the surface of the leg epithelium, from recording #04. The amputation plane is to the right of the frame.

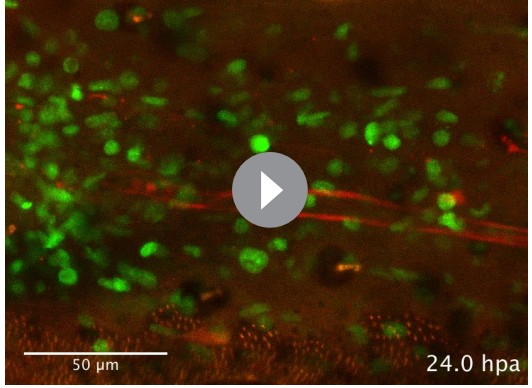

**Video 8.** Live imaging of macrophages engulfing cellular debris in a regenerating *Parhyale* leg. Nuclei are marked by histone-EGFP (*PhHS>H2B-EGFP* transgene, in green) and neurons are marked by DsRed (*DC5>DsRed* transgene, in red). Starting around 35 hr post amputation (hpa), a fragment of DsRed-labelled debris becomes engulfed by macrophages (arrowhead). Maximum projection of multiple focal planes, from recording #05.

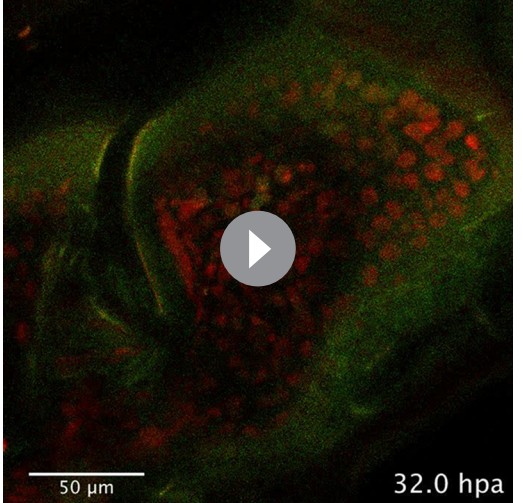

**Video 9.** Live imaging of the onset of cell proliferation in a regenerating leg stump, 32 to 55 hr post amputation (hpa), visualized with a geminin-based cell cycle reporter. Cells in the S, G2 and M phases of the cell cycle are marked by EGFP-PhGemN (*PhHS>EGFP-PhGemN* transgene, in green) and all nuclei are marked by histone-mRuby (*PhHS>H2B-mRuby* transgene, in red) following a heat shock. The loss of the EGFP-PhGemN after mitosis can be seen clearly in a cell marked by an arrowhead. Maximum projection of several focal planes, from recording #13. The amputation plane is at the top-right corner of the frame.

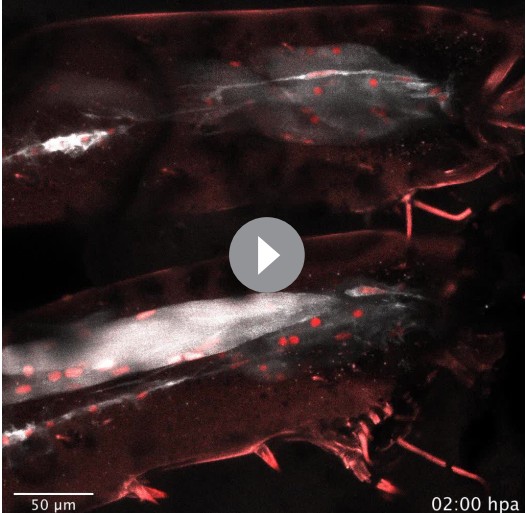

**Video 10.** Live imaging of mesoderm in two regenerating *Parhyale* leg stumps, 3 to 76 hr post amputation (hpa). Two adjacent legs are imaged in a mosaic individual expressing lyn-tdTomato (in white) and H2B-EGFP (in red) specifically in the mesoderm. Muscles at the proximal part of the legs persist, but those near the amputation plane (to the right) degenerate and the inner space of the stump becomes populated by proliferating mesenchymal cells. Maximum projection of multiple focal planes from recording #11. The amputation planes are at the right edge of the frame.

## Acknowledgements

We are grateful to N Konstantinides, A Pavlopoulos, E Kabrani, Z Kontarakis and M Handberg-Thorsager for sharing constructs and transgenic lines; N Konstantinides for advice on live imaging; J Schinko for help with cloning; N Konstantinides, K Echeverri, J Casanova, M Strigini, P Ramos, A Gilles and M Grillo for comments on the manuscript. This work was supported by a Chaire d'Excellence grant from the Agence Nationale de la Recherche (ANR-12-CHEX-0001-01).

## Additional information

### Funding

| Funder | Grant reference number | Author |
| --- | --- | --- |
| Agence Nationale de la Recherche | ANR-12-CHEX-0001-01 | Frederike Alwes<br>Camille Enjolras<br>Michalis Averof |

The funders had no role in study design, data collection and interpretation, or the decision to submit the work for publication.

### Author contributions

FA, Conception and design, Acquisition of data, Analysis and interpretation of data, Drafting or revising the article; CE, Establishment of cell cycle reporter; MA, Conception and design, Analysis and interpretation of data, Drafting or revising the article

### Author ORCIDs

Michalis Averof, http://orcid.org/0000-0002-6803-7251

## Additional files

### Supplementary files

• Supplementary file 1. Sequences for constructs *PhHS>lyn-TdTomato-2A-H2B-EGFP, DC5>DsRed* and *PhHS>EGFP-PhGemN*.

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
