## [Decision Letter]

Congratulations, we are pleased to inform you that your article, "Live imaging reveals the progenitors and cell dynamics of limb regeneration", has been accepted for publication in *eLife*. If you have selected our "Publish on Acceptance" option, your PDF will be published within a few days; if you have opted out of the "Publish on Acceptance" option, your work will be published in about four weeks' time. Please take note of the points below and we hope you will continue to support *eLife* going forwards.

Substantive Comments:

The Reviewers were all very positive and had no major revisions to suggest. This consolidated comment is somewhat repetitive, only to convey the uniformity of opinion. Congratulations.

One Reviewer wrote: In this manuscript Alwes et al. assess the process of limb regeneration in the small crustacean, *Parhyale hawaiensis*, using live recordings over several days, with single cell resolution. The ability to follow the cell dynamics of appendage regeneration using such high resolution imaging over the entire regenerative process is unprecedented, and very powerful. In the article the authors have been able to track cell lineages and follow division planes over long periods of time. They also show that regeneration begins with a variable time of proliferation stasis, followed by a high proliferative phase, associated with the overt morphogenesis of the regenerated new limb. Based on their findings, the authors suggest that limb regeneration in *P. hawaiensis* involves a mixture of epimorphosis and morphallaxis, whereby proliferation occurs and patterning occurs or is finalized after. Although the article is largely descriptive, this work warrants high visibility as it further raises this model organism as an extremely valuable and tractable system to use to investigate the mechanism of appendage regeneration. At present, *P. hawaiensis* remains a significantly underused and under-appreciated model system for regeneration research. This article and many of the others published by the Averof group should raise awareness of this powerful system, so that it becomes more generally adopted and supported by the regeneration community.

The other two had similar views: This is an interesting manuscript. The authors establish live imaging in a new model organism for limb regeneration. The movies are very impressive, documenting prolifertion, movement and fate determination of different cell types. Solid conclusions are drawn about the mechanisms of cell and tissue morphogenesis and differentiation, which will allow rich comparisons across species that can and cannot perform tissue regeneration. Given the importance of stem cell biology and regenerative medicine, a new model system such as this will add tremendous value to the field. This work establishes the use of live imaging to study leg regeneration in *Parhyale*, with single cell detail and high temporal resolution. The methodology developed allowed the authors to describe the regeneration dynamics of multiple cell types in this organ and in particular to demonstrate that there are no specialised progenitor cells involved in regenerating the leg epidermis; most epidermal cells can act as epidermal progenitors. The experiments are carefully designed and the data supports the conclusions. While the work is largely descriptive it is likely to have a significant impact in the field as it shows that *Parhyale* has a huge potential to study regeneration in arthropods and reveals cell and tissue behaviours that have parallels in other species. The technical advances and results presented in this manuscript will stimulate further work that will contribute for the understanding of regeneration not only in *Parhyale* and related arthropods but also in other phyla.